# A measurement framework to assess software maturity models

Reem Alshareef[1,2], Mohammad Alshayeb[1,3], Mahmood Niazi[1,3] and Sajjad Mahmood[1,3]

[1] Information & Computer Science Department, King Fahd University of Petroleum and Minerals, Dhahran, Saudi Arabia
[2] College of Computer Science & Information Technology, Imam Abdulrahman Bin Faisal University, Dammam, Saudi Arabia
[3] Interdisciplinary Research Center for Intelligent Secure Systems, Dhahran, Saudi Arabia



Corresponding author
Mohammad Alshayeb,
Alshayeb@kfupm.edu.sa

## ABSTRACT

Software maturity models can be utilized by organizations to evaluate and enhance their development processes. Established and recognized models such as the Capability Maturity Model Integrated (CMMI) and ISO/IEC 15504 (Software Process Improvement and Capability Determination (SPICE)) have proven their value. However, many new software maturity models exist, and their quality and potential value remain questionable until they are properly assessed before adoption. Without such an assessment, organizations can implement poor or ineffective models, resulting in wasted resources and failed improvement initiatives. Our research aims to address this challenge by developing a measurement framework based on ISO/IEC 15504-3 standards to assess the quality of developed software maturity models. We derived our quality assessment criteria through literature analysis, analyzing four main categories: basic model information, structural design, assessment methods, and implementation support. After developing this framework, we validated it with expert reviews to assess its design and usability and through a series of case studies. Feedback from academics and industry practitioners confirmed the framework's utility, especially recognizing its clear structure and comprehensiveness of evaluation criteria. Case studies also revealed the framework's effectiveness in identifying strengths and areas of improvement, finding that evaluated models had quality scores ranging from 83.3% to 93.2%. Our study enhances software maturity models' practical utility and adoption across different software contexts, providing professionals and academics with a structured way to evaluate and enhance maturity models.

## INTRODUCTION

Organizations often seek ways to improve their practices, and one widely adopted approach is the use of software maturity models. Maturity models provide frameworks for organizations to enhance and continuously improve their processes (*Dahlin, 2020*). The utilization of these models can help companies in outlining the stages or levels of maturity in their software development practices (*Wendler, 2012*). Several capability/maturity models have been adopted as standards for software processes in organizations. Drawing from established engineering and process management principles to help assess and refine

processes by considering process capability and organizational maturity levels (*Röglinger, Pöppelbuß & Becker, 2012*), maturity models can be seen as an effective approach for organizations to enhance their process quality (*Wendler, 2012*; *Pöppelbuß & Röglinger, 2011*). The concept of maturity in software organizations focuses on how systematic processes and practices are implemented. Mature organizations demonstrate this through standardized work processes, ongoing process improvements, clearly defined roles, and management oversight of both product quality and the associated processes (*Weber, Curtis & Chrissis, 1993*). *Moradi-Moghadam, Safari & Maleki (2013)* note an important distinction: mature organizations rely on systematic procedures and established processes, while immature organizations typically achieve results through individual efforts and *ad-hoc* methods.

Software maturity models have been developed across various fields, including engineering (*Tissen et al., 2024*), project management (*Cooke-Davies & Arzymanow, 2003*), and quality assurance (*Al-Qutaish & Abran, 2011*). Organizations such as the International Organization for Standardization (ISO), the International Electrotechnical Commission (IEC), and the Software Engineering Institute (SEI) have created maturity models to assess and improve software processes (*Pöppelbuß & Röglinger, 2011*; *Moradi-Moghadam, Safari & Maleki, 2013*). ISO/IEC 15504, known as Software Process Improvement and Capability Determination (SPICE), stands as one widely adopted standard. This framework helps organizations evaluate software process capability and maturity in a structured manner, assisting them in identifying areas for improvement and maintaining quality development practices. Maturity models serve as guides for determining what improvements to consider and when to implement them (*Weber, Curtis & Chrissis, 1993*). This typically involves evaluating the process against a maturity model. Subsequently, the assessment results are utilized to determine areas where enhancements are needed to strengthen the process maturity level. Models like CMMI and the ISO standard, derived from the SPICE project, have been widely used to enhance software development organizations' processes. Other models, such as the eSourcing Capability Model (eSCMs) and Process Reference Model (PRM), also fall under the software process maturity models (*Dahlin, 2020*). Furthermore, several studies have demonstrated that advancements in maturity within business-specific models positively impact software product quality, project outcomes, and overall productivity (*Cooke-Davies & Arzymanow, 2003*; *Al-Qutaish & Abran, 2011*). Additionally, the use of software maturity models has been linked to enhancements in productivity metrics such as lines of code per staff and increased customer satisfaction (*Bruin et al., 2005*; *von Wangenheim et al., 2010*; *Lasrado, Vatrapu & Andersen, 2015*).

Despite the evolution and widespread use of software maturity models, many available maturity models need more evidence of following empirical methods for developing and validating the models; they are merely conceptual. Therefore, several researchers have emphasized the need for additional empirical validations (*Lasrado, Vatrapu & Andersen, 2015*; *Lahrmann et al., 2011*; *Lasrado et al., 2016*). *dos Santos-Neto & Costa (2019)* conducted an extensive systematic literature review examining 409 articles published between 1973 and 2017. They revealed that only 12 articles addressing the maturity models

discussed in the literature had been validated. This highlights a critical gap, as only 3% of the articles addressed validation. In line with this work, Wendler criticized the scarcity of validation studies (*Wendler, 2012*). Established models like CMMI and SPICE have demonstrated their value through extensive industry implementation; on the other hand, newer domain-specific software maturity models often lack sufficient validation before deployment. The importance of validating software maturity models is underscored by the principle that they should be usable, complete, and accurate within their intended scope (*Bruin et al., 2005*). Newly proposed models, however, are not adequately validated. Robust validation can be done through case studies, surveys, interviews, and focus group discussions (*Bruin et al., 2005*). Additionally, *Rosemann & Vessey (2008)* suggest checks of applicability with practitioners, as gathering varied viewpoints and insights from practitioners in the field aims to enhance the model with a comprehensive spectrum of knowledge and experiences. Nevertheless, such a process is rather burdensome and requires significant time and effort. This lack of validation highlights the importance of a formal method to evaluate the quality and efficacy of software maturity models. In the absence of such a framework, organizations risk implementing models that may be inadequate or ineffective, leading to wasted resources and failed improvement initiatives.

The main objective of this work is to develop a framework that can evaluate software maturity models against a set of criteria. The results of the assessment process identify the strengths and weaknesses of each evaluated model. We begin by conducting a literature review to identify existing frameworks and success factors in maturity model development. We then empirically verify and validate the framework through expert evaluations, engaging industry practitioners and academics to assess the framework's design and effectiveness. We further reinforce this validation with evidence from multiple case studies by applying the framework to evaluate different types of maturity models across various domains. Through literature analysis, expert evaluations, and case studies, we aim to ensure that our framework is theoretically sound and practically applicable. The entire study is conducted following ISO/IEC 15504-3 standard guidelines to ensure that the results of the evaluations and assessments are reliable and comparable. Through this methodologically rigorous approach, we aim to provide a validated and practical method for assessing software maturity models' quality.

## LITERATURE REVIEW

Creating, applying, and evaluating software maturity models in the software engineering environment are essential as these activities help guide organizations toward better software development practices. The first activity we undertook was to survey the literature in the following domains:

- Software process maturity models
- Software product maturity models
- Design principles for developing maturity models
- Available assessment frameworks

We only considered academic sources from conferences and peer-reviewed journals that focused on the creation of software maturity models, as well as standards and technical documentation from reputable organizations such as ISO/IEC and well-known industry frameworks. The sources had to be directly related to software maturity models, use rigorous methodological techniques such as empirical validation studies, and come from credible sources.

## Software process maturity models

Several maturity models have been created following the structure and principles of CMMI. Examples of these models include:

- Software Engineering Maturity Model (SEMM) (*Garzás et al., 2013*): The AENOR workgroup has proposed the Software Engineering Maturity Model. SEMM is a process maturity model based on the ISO/IEC 12207 standard (*ISO, 2008*). The main goal of this model is to assess the maturity of software processes in small enterprises by examining capacity levels corresponding to different maturity levels.
- IT Service Management Maturity Model (*Picard, Renault & Barafort, 2015*): *Picard, Renault & Barafort (2015)* introduced the IT Service Management maturity model. This model is intended to evaluate IT infrastructure processes and aligns with ISO/IEC 20000-1 standards (*ISO, 2011*). Organizations can adopt this model to evaluate and enhance their IT service management capabilities, leading to improved service delivery.
- Test Maturity Model Integration (TMMI): The TMMI model, introduced by the TMMI Foundation (*van Veenendaal & Wells, 2012*), aims to enhance testing processes within organizations (*Garousi & van Veenendaal, 2022*). The authors suggest that TMMI complements CMMI and focuses on improving the maturity of the testing process, which they argue is not adequately addressed by CMMI.
- Software Process Improvement and Capability Determination (SPICE) (*Dorling, 1993*): ISO/IEC has proposed this framework, which draws on ISO/IEC 12207 (*ISO, 2008*). SPICE enhances processes within software organizations and assesses their capacity to deliver high-quality software products.

## Software product maturity models

Product maturity models aim to define and measure the maturity of software products. Below are some of the prominent ones:

- Software Product Quality Maturity Model (SPQMM): The Software Product Quality Maturity Model was proposed by *Al-Qutaish & Abran (2011)* to assess software product quality. Initially, it calculates quality levels based on characteristics, sub-characteristics, and measurements derived from ISO 9126 (*ISO/IEC, 1991*). These values are then combined into a single value, converted into a six sigma value, and the software's integrity level is determined using ISO 15026. Finally, the overall maturity level is identified. Assessors are required to use ISO 9126 quality attributes and metrics, making SPQMM dependent on this specific standard.

- SCOPE: Developed by the EuroScope consortium (*Jakobsen, O'Duffy & Punter, 1999*), the SCOPE product maturity model (SMM) evaluates software product maturity across five levels. Levels 2, 3, and 4 incorporate standards such as ISO 12119, ISO/IEC 9126, and ISO 14598. SMM evaluates quality by matching the specifications with the specified requirements. Additionally, SMM requires documentation of the process to ensure that the final product meets specifications. However, SMM does not focus on the final product's quality (*i.e.*, the code).

- Open-Source Maturity Model (OSMM): This model was developed by *Golden (2005)* to assist organizations in effectively implementing open-source software through a three-phase process. The process involves assessing maturity elements by defining requirements, locating resources, evaluating element maturity, and assigning scores. The second phase involves applying weighting factors, and the third phase involves calculating the overall product maturity score. Designed to be lightweight, OSMM can evaluate an open-source product's maturity within 2 weeks.

- Technical capability model (TCMMI): TCMMI (*Abdellatif et al., 2019*) evaluates the maturity of a software product as an indicator of its quality. The framework is divided into a reference model and an assessment method. The reference model offers a structure for collecting product quality indicators as evidence of the product's capability and maturity. The assessment method follows standard procedures to evaluate the product's maturity by measuring its conformance to the quality attributes defined and agreed upon by the product's stakeholders.

## Design principles for developing maturity models

Examining the various widely used maturity models reveals that the process of model design and the specific steps followed in the design are not explicitly detailed (*Becker, Knackstedt & Pöppelbuß, 2009*). This raises an issue as the research steps cannot be followed in terms of repeatability, verifiability, and completeness (*Frick, Küttner & Schubert, 2013*). Searching the literature, we will explore significant contributions in this area, looking at previous attempts to propose methodologies and specific guidelines for creating maturity models. Various studies that revolved around creating new maturity models have attempted to develop guidelines to standardize the development of maturity models. These efforts involved standardizing vocabulary use and adhering to certain validated procedures. *Bruin et al. (2005)* developed one of the most important works in this field by proposing a universally applicable framework for creating such models. The authors established six essential phases for model development: scope, design, populate, test, deploy, and maintain. Their work clarifies that following a systematic and phased process is crucial as it ensures the developed model is robust and can be applied in real-world environments. The authors state that creating maturity models is an ongoing process requiring maintenance, regular updates, and adjustments to align with the rapidly changing practices in software development. Additionally, the authors gathered different perspectives and insights from domain experts in an attempt to enrich their framework with a broad range of knowledge and experiences. *Becker, Knackstedt & Pöppelbuß (2009)*

developed a procedural model for creating IT management maturity models, following design science guidelines defined by *Hevner et al. (2004)*. The authors proposed the following steps: comparison with existing models, iterative procedure, evaluation, design as a search process, design as an artifact, research contributions, targeted presentation of results, and scientific documentation. *Otto, Bley & Harst (2020)* proposed a detailed process model to guide users and developers in creating prescriptive and well-validated maturity models following eight steps. Additionally, *Solli-Sæther & Gottschalk (2010)* proposed a detailed process for developing stage models with methodological considerations. Their work presented a framework for developing stage models by theorizing essential topics related to growth stages. Table 1 illustrates the different design principles for developing maturity models.

## Existing assessment frameworks and research gap

Only one assessment method that is related to maturity models was found during our review of the existing maturity models. A framework for evaluating digital health maturity models was presented by *Woods et al. (2022)* and offers a way to compare the various models that are currently available. The authors acknowledge the importance of evaluating digital health maturity models since they can eventually help achieve better healthcare outcomes. The framework has a total of four distinct categories: actionability, completeness, feasibility, and healthcare context assessments. The healthcare context assessment category assesses the suitability, practicality, and relevance of the model to health settings. The feasibility assessment category assesses the vendor's commitment to improvements, access to data, the requirements to implement the model, and the resources required. The completeness assessment category assesses how the model expresses maturity across seven dimensions and 27 indicators. The actionability assessment category assesses whether the model's recommendations are actionable, feasible, and consistent with the organization's goals. Our goal of developing a framework to evaluate software maturity models can be achieved by following the structured approach detailed in the work of *Woods et al. (2022)*. Their focus on context, feasibility, completeness, and actionability provides us with an evaluation approach that can be adapted for our framework (*Woods et al., 2022*).

Although extensive research exists around maturity model creation and the widely available established guidelines, a notable gap can be observed by examining the current literature. Missing from this body of work is a unified evaluation approach. Current literature fails to present an integrative assessment methodology that combines multiple dimensions of software maturity model development into a single assessment framework. To the best of our knowledge, no framework in the literature has focused on evaluating the quality of software maturity models. The absence of a comprehensive assessment framework clearly supports the opportunity for developing an integrative assessment and measurement framework that provides a holistic view of the software maturity model's overall quality. Assessing software maturity models is critical as it eventually leads organizations to utilize validated models to improve and evolve their software development practices. To bridge this identified gap, we aim to develop an assessment framework for software maturity models that can be both adaptable and comprehensive.

**Table 1 The different design principles for developing maturity models.**

| Author | Purpose | Design principles | Key contributions |
|---|---|---|---|
| Bruin et al. (2005) | A framework for creating maturity models relying on sequential phases. | The methodology is based on principles that emphasize the importance of a structured, iterative process in developing maturity models. | The model is based on sequential phases:<br>• Scope<br>• Design<br>• Populate<br>• Test<br>• Deploy<br>• Maintain |
| Becker, Knackstedt & Pöppelbuß (2009) | A procedural model for creating IT management maturity models. | Design science research (DSR). | Introduced a comprehensive approach to maturity model development based on the guidelines for design science defined by Hevner et al. (2004):<br>• Comparison with existing models.<br>• Iterative procedure.<br>• Evaluation.<br>• Design as a search process.<br>• Design as an artifact.<br>• Research contributions.<br>• Targeted presentation of results.<br>• Scientific documentation. |
| Otto, Bley & Harst (2020) | A detailed process model to guide users and developers in creating prescriptive and well-validated maturity models. | Design-oriented approach. | A detailed process model for creating maturity models structured around eight steps:<br>• Define the problem and scope.<br>• Understand the domain.<br>• Identify the need for a new model.<br>• Define levels and dimensions.<br>• Shift to a prescriptive maturity model.<br>• Evaluate the final draft.<br>• Apply the model in a real-world setting.<br>• Document the final maturity model. |
| Solli-Sæther & Gottschalk (2010) | A detailed process for developing stage models with methodological considerations. | Goal-oriented procedure | Develop the following procedure for the stages of the growth modeling process:<br>• Suggested Stage Model<br>• Conceptual Model<br>• Theoretical Model<br>• Empirical Model<br>• Revised Stage Model |
| Mettler (2011) | A phase model for developing and applying maturity models that address theoretical soundness issues. | Design science research (DSR). | A detailed four-phase development cycle for maturity assessment models that includes the following:<br>• Define Scope: Setting the model's breadth and analysis level.<br>• Design Model: Creating the model based on a clear definition of 'maturity'.<br>• Evaluate Design: Verify and validate the model.<br>• Reflect Evolution: Considering how changes can be made and the evolution over time. |
| Dikhanbayeva et al. (2020) | Assess industry 4.0 maturity models based on core design principles. | A set of measurable and attainable steps for development according to Industry 4.0 design principles. | A detailed analysis of 12 maturity models against 8 core Industry 4.0 principles. Identifying gaps for future research and development strategy. |

 

Our assessment framework will incorporate different criteria and provide a solid approach for assessing and improving software maturity models. Filling this gap will contribute to advancing the development of maturity models in the field. This framework will help navigate the complexity of evaluating and validating software maturity models, improving the software and process quality.

# SOFTWARE MATURITY MODELS ASSESSMENT FRAMEWORK

In this section, we discuss the research methodology that we adopted in developing our assessment framework.

## Research methodology

Our research methodology consisted of five phases, as shown in Fig. 1:

(1) Literature review: Initially, we conducted a review of existing literature to understand the characteristics of software maturity models and what indicates a successful model. To do this, we reviewed several articles and industry reports to acquire insights on the strengths and weaknesses of current software maturity models. For instance, the work conducted by *Picard, Renault & Barafort (2015)* and *Otto, Bley & Harst (2020)* revealed some important factors that can be used in evaluating software maturity models.

(2) Framework design and development: Based on the literature review, we created an assessment framework that involved the elicited evaluation criteria. The framework consists of the key aspects that we found important in developing high-quality maturity models. For instance, we considered aspects such as scalability, relevance, empirical evidence, and practical usability.

(3) ISO/IEC TR 15504-3 integration: We reviewed the ISO/IEC TR 15504-3 assessment process guidelines to incorporate relevant features into our measurement framework to aid the assessment process.

(4) Validation: After developing our criteria, we conducted a pilot study to evaluate several available software maturity models to assess how effectively our criteria capture maturity model quality. This pilot was carried out by two domain experts who reviewed the criteria in depth and suggested several refinements, including merging overlapping items and removing less relevant ones. Based on their feedback, we adjusted the criteria to reflect the essential aspects of quality more accurately. Once all researchers reached a consensus on the completeness and clarity of the revised criteria, we gathered structured feedback from academics and practitioners. The feedback concentrated on the participants' overall satisfaction with the framework, its overall structure, its assessment capabilities, and its ease of use. We conducted four detailed case studies; participants were selected through purposive sampling, inviting academics who had developed maturity models in different domains. Each participant assessed their own developed model using our framework under researcher guidance to ensure objectivity. These case studies provided comprehensive empirical validation of our framework's effectiveness across diverse contexts, helped us identify areas for

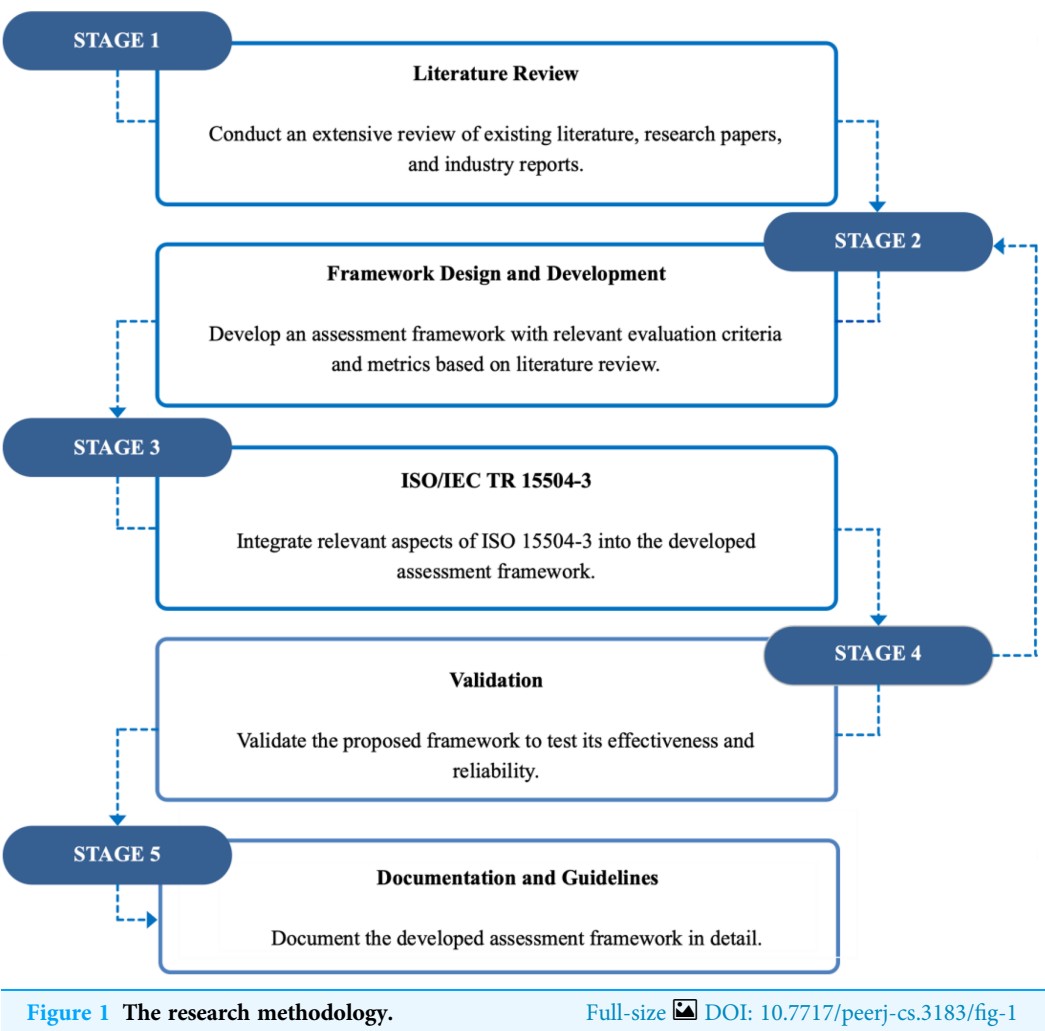

Figure 1 The research methodology.

improvement, and provided us with insightful information about our developed framework.

(5) Documentation and guidelines: We present our developed assessment framework and provide instructions and supporting materials to assist users in implementing it.

## The developed measurement framework

We utilized a focused literature search strategy based on the most reliable, established, and methodologically sound works in maturity model development and evaluation. Instead of reviewing all existing maturity model publications comprehensively, we aimed to strategically select foundational and influential sources that represent state-of-the-art maturity models' assessment and development principles. The literature selection criteria were based on identifying the works that significantly contributed to the maturity model theory, methodology, and assessment practices. Sources were further assessed to select those that demonstrated impact, methodological rigor, and comprehensive coverage of

assessment principles. We carefully considered frameworks and standards such as ISO/IEC (SPICE) and Capability Maturity Model Integration (CMMI) version 2.0 in this process. These well-known models were used to understand the key aspects of maturity and the stages of maturity levels. The identified evaluation criteria capture aspects of the model's development and design. These criteria address design elements, such as clarity, standardization, documentation, and development aspects, including accuracy, effectiveness, validation, practical implementation, and ongoing support (*dos Santos-Neto & Costa, 2019*; *Maier, Moultrie & Clarkson, 2012*). Furthermore, insights gained from academic articles such as the work done by *Bruin et al. (2005)* and *Wendler (2012)* were instrumental in extracting several criteria for developing effective maturity models. Their emphasis on developing maturity models based on sequential phases of development (*e.g.*, scope, design, evaluation) influenced the development of criteria such as criterion 1.2.

We aimed to capture comprehensive criteria across scalability, empirical evidence, and usability in practice. These themes consistently emerged in the literature as essential to high-quality maturity models. We drew upon the work of *Frick, Küttner & Schubert (2013)* for the need for tool support and *Bruin et al. (2005)* for practical implementation in real-world scenarios. Our criteria also reflected this since it indicates that support and real-world application are important in developing a robust maturity model.

The framework development was guided by a comprehensive literature review complemented by insights derived from the authors' prior work in developing and implementing multiple software maturity models (*Niazi et al., 2020*; *Alam et al., 2024*). Combining literature review and domain-specific expertise enabled a comprehensive approach to criteria formulation. Figure 2 illustrates our framework development methodology, showing the clear progression from three primary input sources (literature review, existing maturity models, and authors' prior knowledge) through systematic criteria identification and categorization, ultimately resulting in our comprehensive assessment framework. The assessment criteria were grouped into four categories: model basic information, model structure, model assessment, and model support. Each of these categories is important for developing and applying software maturity models. The categories were derived through thematic analysis of our literature review findings. This categorization emerged from identifying recurring themes and critical success factors across the examined sources. This four-category structure captures several aspects of maturity model development, from conceptual foundation through practical application, ensuring comprehensive quality assessment across all critical dimensions. Furthermore, an automated assessment tool is available at https://zenodo.org/records/15834005, allowing users to evaluate models more efficiently. Each criterion is assessed using a 5-point scale:

0–No

1–Somewhat/Maybe

2–Yes

U–Unknown

NA–Not Applicable

Scores are summed within each category and converted to percentages. We applied equal weighting to all evaluation criteria to ensure fairness, simplicity, and transparency.

Alshareef et al. (2025), *PeerJ Comput. Sci.*, DOI 10.7717/peerj-cs.3183

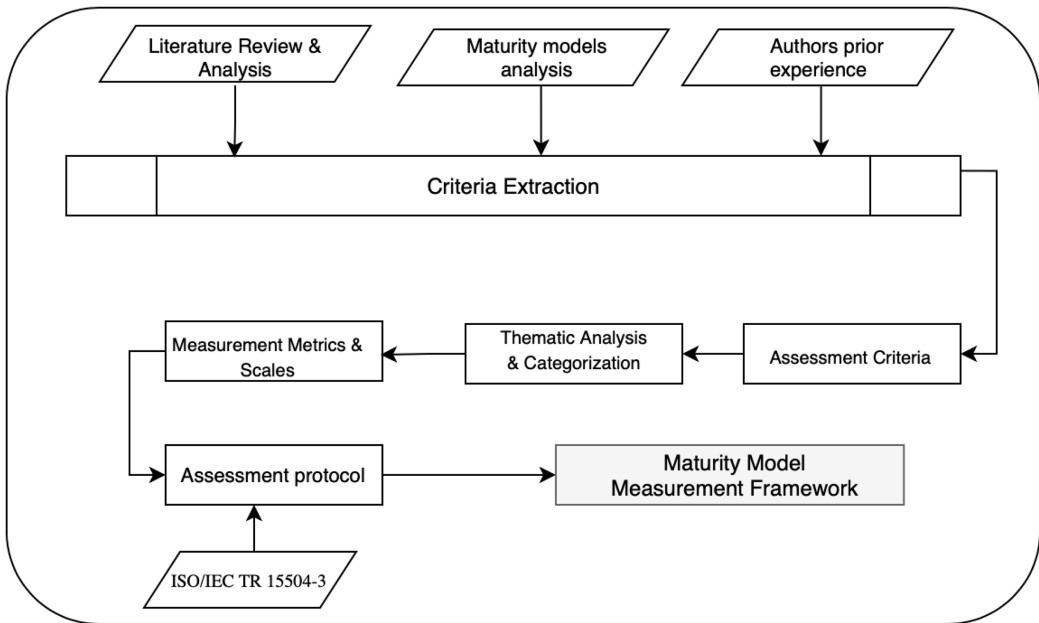

**Figure 2 The framework development methodology.**

| Table 2 The specific software development contextsin which the maturity model can be applied. | |
| --- | --- |
| **What software maturity model is being assessed?** | |
| **What is the size of the organization that the maturity model is designed for?** | Yes |
| Small organizations (10–50 employees) | ☐ |
| Medium organizations (51–249 employees) | ☐ |
| Large organizations (250 + employees) | ☐ |
| **What is the main focus of this maturity model?** | Yes |
| Web development | ☐ |
| Mobile development | ☐ |
| Enterprise software | ☐ |
| Other | ☐ |
| **If other, specify:……………….………………….………..** | |

Since there is no established evidence or consensus on which criteria are more important, treating each equally avoids introducing subjective bias. This approach also makes the evaluation process easier to understand and replicate.

Table 2 identifies the specific software development contexts in which the maturity model can be applied. Each of the four categories is detailed below:

- **The model's basic information:** This category covers the basic aspects that define the purpose of a software maturity model. It includes the theories or frameworks on which the model is built and the application domain. Additionally, this category emphasizes how the model differs from others and offers unique value. Table 3 contains the criteria for this category.

**Table 3  The criteria for model's basic information.**

| 1 | Model basic information | Score |
|---|---|---|
| | | 0  1  2  U  NA |
| 1.1 | Are the costs of implementing this maturity model (*e.g.*, initial, implementation, recurring fees) reasonable relative to the value of applying it? | |
| 1.2 | Were predefined steps followed in the research design of the maturity model? | |
| 1.3 | Is the maturity model distinct and unique compared to existing models in the same domain? | |
| 1.4 | Does the maturity model conform to established industry standards or guidelines? | |
| 1.5 | Does the model clearly specify its relevance to particular domains or areas? | |
| 1.6 | Has the model been validated in real-world settings (*e.g.*, peer-reviewed literature, surveys, industry groups), demonstrating its applicability and effectiveness? | |
| 1.7 | Does the model have a clear ideological foundation supported by established theories or models? | |
| 1.8 | Is the model evidence-based (*e.g.*, grounded in the peer-reviewed literature, industry-recognized best practice)? | |
| 1.9 | Are the model's practices applicable across different scenarios, cases, and projects? | |
| **SECTION SUBTOTAL** | | |

**Table 4  The criteria for the model's structure.**

| 2 | Model structure criteria | Score |
|---|---|---|
| | | 0  1  2  U  NA |
| 2.1 | Is the process of applying the model clear? | |
| 2.2 | Does the model provide clear definitions of maturity and dimensions of maturity? | |
| 2.3 | Are maturity levels within the model clearly defined, with each level described by clear criteria and expected outcomes? | |
| 2.4 | Does the maturity model outline specific levels and the logical progression between these levels? | |
| 2.5 | Is the maturity model's structure, including the number of levels and attributes, clear and coherent? | |
| 2.6 | Does the model propose specific improvement measures or practices for advancing from one maturity level to the next? | |
| 2.7 | Is there an ability to adjust or alter the model's structure, components, or processes (*e.g.*, the model can evolve and remain relevant.)? | |
| 2.8 | Is there a balance in the model between reflecting the complexities of the domain and maintaining simplicity for understandability? | |
| 2.9 | Is the maturity model's constructs and definitions accurate and precise? | |
| 2.10 | Is the maturity model easily accessible and usable by practitioners without extensive training? | |
| **SECTION SUBTOTAL** | | |

- **Model structure:** Table 4 outlines the key criteria for the model's structure, focusing on its architecture and design. This includes the various maturity levels and elements illustrating different maturity aspects. An important aspect that is addressed in this category is the necessity for clear and straightforward definitions, as these help organizations accurately assess their current maturity level. Additionally, it addresses criteria that emphasize the simplicity and user friendliness of the software maturity model being evaluated.

- **Model assessment:** The assessment category's criteria focus on the methods utilized by organizations to determine their maturity levels. This involves evaluating the methods used for assessments, ensuring the reliability of the tools, and validating of the

**Table 5  The criteria for the model's assessment.**

| 3 | Model assessment criteria | Score | | | | |
|---|---|---|---|---|---|---|
| | | 0 | 1 | 2 | U | NA |
| 3.1 | Were the model's assessment instruments validated to ensure accuracy and reliability? | | | | | |
| 3.2 | Are there clear, precise criteria for assessing maturity at each level and dimension, allowing for consistent and objective evaluations? | | | | | |
| 3.3 | Does the model include a detailed methodology for conducting assessments, providing guidance on evaluating criteria, and interpreting results? | | | | | |
| 3.4 | Does the assessment methodology outline clear procedures for assessors? | | | | | |
| 3.5 | Is there a logical connection between the model's design and the chosen assessment methods? | | | | | |
| 3.6 | Does the model support different types of assessments (*e.g.*, self-assessment, third-party assessment)? | | | | | |
| 3.7 | Can support be provided during the assessment using the model? | | | | | |
| 3.8 | Does the maturity model promote transparency and openness in identifying and addressing areas for improvement, including the possibility of suggesting enhancements? | | | | | |
| 3.9 | Does the maturity model leverage technology and tools for more efficient and accurate assessments? | | | | | |
| **SECTION SUBTOTAL** | | | | | | |

assessment instruments. These criteria help guarantee that evaluations are objective and evidence-based, enabling organizations to measure their progress accurately. Also, the availability of various assessment methods is considered, for example, self-assessments or third-party assessments, which may be warranted for a variety of needs and situations. The criteria for this category are shown in Table 5.

- **Model support:** This category focuses on the availability of support mechanisms essential for the effective implementation of the software maturity model. It includes criteria such as training, documentation, and guidance. These resources help organizations apply the model to their specific needs and help them understand the results clearly. Furthermore, it highlights the importance of keeping the model relevant and regularly updated to maintain its applicability over time. Table 6 includes the criteria for this category.

## Results summary

To ensure we can appropriately and objectively evaluate different maturity models, we should note the difference between the total and the grand total. the total represents the raw sum of valid scores across all applicable criteria, excluding items marked as not applicable (NA) or unknown (U). In contrast, the grand total is a normalized score calculated using the formula:

$$\left( \frac{\textbf{TOTAL}}{\textit{Total number of applicable criteria } \times \textbf{ 2}} \right) \times 100$$

Therefore, the model is not penalized for criteria that involve something outside its intended scope or are insufficiently defined in the source material. By excluding NA and U scores from the denominator, the grand total reflects only the evaluated dimensions, thus

**Table 6 The criteria for the model's support.**

| 4 | Model support criteria | Score |
|---|---|---|
| | | 0 1 2 U NA |
| 4.1 | Does the maturity report present the results clearly? | |
| 4.2 | Is there adequate documentation supporting the application of the assessment, such as a handbook, textual descriptions, or software assessment tools? | |
| 4.3 | Is the model designed with enough flexibility to be adapted to different organizational settings? | |
| 4.4 | Does the model provide actionable insights and guidance for both practitioners and researchers? | |
| 4.5 | Does the maturity report provide practical, useful recommendations to drive improvements? | |
| 4.6 | Does the model facilitate benchmarking against industry standards or comparisons with similar organizations? | |
| 4.7 | Can the maturity report be customized? | |
| 4.8 | Is training available for effectively implementing and utilizing the maturity model? | |
| 4.9 | Is there a continuity and evolution plan between different versions of the model with accessible documentation? | |
| 4.10 | Is there a maintenance plan in place to ensure the model remains relevant and up-to-date? | |
| **SECTION SUBTOTAL** | | |

**Table 7 The assessment results per category.**

| Section | Result |
|---|---|
| Basic information | |
| Model structure criteria | |
| Model assessment criteria | |
| Model support criteria | |
| Total[1] | |
| Grand total[2] | 0% |

Notes:
[1] The sum of valid scores across all categories.
[2] $\left( \frac{TOTAL}{Total\ number\ of\ applicable\ criteria \times 2} \right) \times 100$.

preserving the integrity and comparability of the assessment outcomes. Table 7 illustrates the assessment results per category.

## Score interpretation

**0–25:** Basic—The software maturity model is in the early stages, requiring significant development.

**26–50:** Emerging—The software maturity model shows foundational strengths but needs further refinement for broader applicability and impact.

**51–75:** Mature—The evaluated criteria demonstrate a functional approach, suggesting a reasonably developed software maturity model.

**76–100:** Advanced—The model meets standard maturity expectations, demonstrating broad applicability and significant impact.

Literature analysis revealed common practices in maturity assessment, where a four-level categorizations are widely used, similar to CMMI's initial, managed, defined, and optimizing levels. Expert consultations with three experts confirmed that these ranges correspond with practical expectations for quality assessment of maturity models. The ranges were validated through pilot testing with two maturity models to ensure that the categorizations effectively differentiate between models of varying quality levels.

## ALIGNMENT WITH ISO/IEC TR 15504-3 STANDARD

Our approach to assessing software maturity models is guided by the ISO/IEC TR 15504-3 standard, which outlines clear and structured practices for software process assessments. Using this framework helps ensure that assessments are consistent, well-balanced, and reliable. Recognizing that ISO/IEC TR 15504 would enable us to meet our research objectives for the previously listed reasons, further considerations influenced our decision to select ISO/IEC TR 15504-3. First, it is an internationally recognised standard for process assessment methodology, which provides credible and validated information on systematic evaluation approaches. Second, the standard specifically addresses assessment methodology, assessment procedures, and quality requirements, all of which relate closely to our objective of developing a rigorous assessment framework for software maturity models. Finally, the standard was intended to be applicable to multiple process domains and organizational contexts, making it particularly suitable for assessing software maturity models that span diverse software development approaches and organizational structures. The assessment method is divided into three phases:

- Input phase: This phase determines assessment requirements, including purpose, scope, constraints, and resources. The result is a clear set of parameters for the assessment process.
- Processes phase: This phase consists of the actual assessment process, including data collection, analysis, validation, and scoring against predefined criteria, using standardized procedures.
- Output phase: This phase consists of documenting, reporting, and communicating assessment results, including detailed justification for decisions and recommendations for improvement.

To ensure assessment consistency, all evaluators used defined standardized guidelines in accordance with ISO/IEC 15504-3 guidelines. The reliability of the assessments is supported by the structured evaluations and oversight from the researchers. Each phase is discussed separately below.

### Defining the assessment input

Our approach begins by defining the assessment input, which includes the purpose, scope, and any constraints that may affect execution. We clearly specify the aspects of the models

to be assessed, including their impact on enhancing software development processes and their relevance for various sizes and types of organizations. The assessment input will be collected during the initial phase, following these steps:

- Step 1: The developer of the maturity model prepares the maturity model to be assessed.
- Step 2: The developer of the maturity model appoints themselves or another qualified individual to be responsible for the assessment process of the maturity model.
- Step 3: In consultation with domain experts, the developer selects qualified assessors who will participate in the assessment.
- Step 4: The developer or the assessor defines the scope and objectives of the assessment:

  ○ Scope: To evaluate the effectiveness and comprehensiveness of the maturity model.
  ○ Objectives: Identify strengths, weaknesses, and areas for improvement in the maturity model.

- Step 5: The developer or the assessor lists the various artifacts of the developed software maturity model to be evaluated.
- Step 6: The developer ensures that the assessor has access to all required documents needed for the assessment.
- Step 7: The developer or the assessor defines a clear schedule for the assessment process.
- Step 8: The developer ensures the assessor is available during the assessment period.

Any changes to the assessment input should be approved and documented by the maturity model developer and assessors.

## Structured assessment process

As described above, the framework has four main categories, each evaluated against a set of predefined criteria. These criteria reflect the software maturity models' efficiency, relevance, and thoroughness. Once the assessment input from the previous phase has been finalized, the assessment process moves to the next phase with the following steps:

- Step 1: The developer confirms that all necessary documents are available and accessible for the assessment.
- Step 2: The developer or the assessor validates the consistency and sufficiency of the collected data before proceeding with the assessment.
- Step 3: The developer or the assessor assigns a score to each criterion in the defined categories.
- Step 4: The developer or the assessor calculates each category's scores by summing each criterion's score.
- Step 5: Convert the total score to a percentage (out of 100).
- Step 6: Based on the overall score, determine the quality of the maturity model.

## Assessment output

After completing each phase, it's important to document and report the findings obtained during that phase in detail as part of the output process. The developer of the maturity model is responsible for recording the assessment results in a comprehensive document. The resulting document provides detailed explanations for each decision made and the scores given, thus ensuring all supporting evidence is documented.

## FRAMEWORK EVALUATION

In this section, we describe how we evaluated the effectiveness and feasibility of our measurement framework. We aimed to elicit domain experts' feedback on our proposed framework. Before the evaluation, we provided the experts with a complete description of the framework and usage guidelines. The evaluation criteria for the framework included:

- Ease of use: Is the framework simple and intuitive, enabling users with varying levels of expertise to use it without extensive training?
- Framework structure: Is the framework's structure comprehensively designed to cover all critical elements necessary to thoroughly assess software maturity models, ensuring a holistic evaluation approach?
- Assessment: Does the framework provide robust and reliable metrics for evaluating the strengths and weaknesses of software maturity models, ensuring detailed analysis and actionable insights?
- User satisfaction: Does the framework support user satisfaction by evaluating its utility, ease of use, and overall impact on improving software development practices across diverse environments?

For each of the criteria outlined above, at least two assessment points were created and evaluated using a five-point Likert scale that ranged from Strongly Agree to Strongly Disagree. Expert participants were selected using purposive sampling based on the following criteria. First, professional relevance was essential, requiring direct experience in software engineering, software development, or related technical roles. Second, participants were required to have an appropriate educational background with a minimum of graduate-level education in software engineering, computer science, or related fields. Finally, domain expertise was necessary, encompassing knowledge of software maturity models, process assessment methodologies, and software quality frameworks. This approach resulted in a diverse expert panel of eleven participants comprising academic researchers, PhD candidates, software engineers, and a cybersecurity analyst, with experience ranging from under 3 years to over 10 years in relevant domains, as detailed in Table 8.

## Expert feedback analysis and discussion

The integration of expert feedback served as a critical mechanism for ensuring our measurement framework's relevance and effectiveness. We analyzed the survey results

**Table 8 The expert panel profiles.**

| | | Years of experience in software assessment or process improvement | | | | |
|---|---|---|---|---|---|---|
| | | Less than 3 years | 3 to 5 years | 6 to 10 years | More than 10 years | Total |
| **Current role** | Academic researcher | 0 | 1 | 1 | 2 | 4 |
| | Project manager | 0 | 0 | 0 | 0 | 0 |
| | Software engineer | 1 | 0 | 0 | 0 | 1 |
| | PhD candidate | 2 | 2 | 0 | 1 | 5 |
| | Other | 1 | 0 | 0 | 0 | 1 |
| | **Total** | 4 | 3 | 1 | 3 | 11 |

from eleven domain experts, with diverse experience levels in software process assessment, revealing several critical findings.

The evaluation demonstrated strong positive outcomes. A total of 81.8% of experts agreed, and 18.2% strongly agreed that the framework is logically structured and easily understandable, indicating excellent foundational design acceptance. Furthermore, 90.9% of experts affirmed that the criteria are self-explanatory, with only one expert expressing concerns. This demonstrates the framework's clarity and accessibility for users with varying expertise levels. Expert validation of the framework's evaluation capabilities showed positive results, with 72.7% agreeing and 27.3% strongly agreeing that it provides precise and reliable indicators for identifying areas of weakness and strength in software maturity models. This confirms the framework's ability to provide meaningful and accurate evaluations. The agreement on the framework's time efficiency is particularly noteworthy, with all eleven experts agreeing that the assessment process is time-efficient and resource-conservative. This addresses a critical practical concern for framework adoption in industry settings.

The framework's comprehensive coverage received strong validation, with 81.8% agreeing and 18.2% strongly agreeing that it is thorough and considers all critical elements vital for assessing software maturity models. Additionally, 90.9% of experts confirmed the framework's applicability across diverse software development contexts and organizational sizes, demonstrating its flexibility and broad utility.

The recommendations criterion produced similarly strong results, with 81.8% agreeing and 18.2% strongly agreeing to recommend the framework to colleagues and industry peers as a useful evaluation tool. We had balanced results when asking about the need for training before utilizing the framework, with 27.3% disagreeing, 27.3% being neutral, and 45.5% agreeing that some training is required. The results indicate that additional training may enhance users' confidence, ultimately improving our assessment framework's effectiveness. We also plan to supplement our framework with further guidance materials to increase the practicality of the assessment process. Additionally, to further facilitate users' understanding of the criteria, several criteria have been reworded for clarity and comprehensibility. Table 9 illustrates evaluation scores and expert responses for each criterion.

**Table 9 The evaluation scores and expert responses for each criterion.**

| | No. of participants = 11 | | | | |
| | Positive | | Negative | | Neutral |
| | SA | A | SD | D | N |
| --- | --- | --- | --- | --- | --- |
| **Framework ease of use** | | | | | |
| The framework is logically structured and understandable by academic researchers and industry practitioners. | 2 | 9 | 0 | 0 | 0 |
| All the criteria in the framework are self-explanatory and require no further clarification. | 4 | 6 | 0 | 1 | 0 |
| The framework is easy to learn, enabling individuals with varying levels of expertise in software maturity models to utilize it effectively. | 3 | 7 | 0 | 0 | 1 |
| Some kind of training is necessary for the utilization of this framework | 0 | 5 | 0 | 3 | 3 |
| **Framework structure** | | | | | |
| The framework is thorough and considers all critical elements vital for assessing software maturity models. | 2 | 9 | 0 | 0 | 0 |
| The assessment process defined by the framework is time-efficient and resource-conservative. | 0 | 11 | 0 | 0 | 0 |
| **Framework assessment capabilities** | | | | | |
| The framework provides precise and reliable indicators for assessing the quality of software maturity models. | 3 | 8 | 0 | 0 | 0 |
| The framework can identify areas of weakness and areas of strength in a software maturity model. | 3 | 8 | 0 | 0 | 0 |
| **User satisfaction** | | | | | |
| I recommend this framework to colleagues and industry peers as a reliable tool for evaluating software maturity models. | 2 | 9 | 0 | 0 | 0 |
| The framework is applicable across diverse software development contexts and organizational sizes. | 1 | 9 | 0 | 0 | 1 |

# DOMAIN-SPECIFIC FRAMEWORK

The following section serves as a guide for customizing the evaluation criteria for a given software domain. A domain-specific framework is an adaptation of the original assessment framework used to evaluate maturity models in specific software domains (*e.g.*, security, quality, outsourcing). These customized frameworks are adjusted to reflect the target domain's characteristics, standards, and requirements. This includes modifying the evaluation criteria to address aspects applicable to a specific domain, to ensure that the maturity model accurately assesses and supports improvements relevant to that specific context. We outline a systematic approach to customize the criteria and develop a domain-specific framework through the following steps:

- **Step 1: Determine unique domain requirements**: The user should identify the aspects unique to the specific domain under consideration (*e.g.*, security, quality, outsourcing, *etc.*). This includes standards, practices, and concerns specific to the domain. For example, one of the security domain requirements is compliance with security standards (*e.g.*, ISO/IEC 27001). Another example is the relevance to specific security domains such as cybersecurity, information security, and incident response.
- **Step 2: Map domain requirements to existing criteria**: For each criterion in the software maturity model, determine how it can be adapted to fit the unique requirements of the chosen domain. This involves revising the language and focus of each criterion to address domain-specific aspects identified in Step 1. For example, for the first elicited domain requirement, align the need for security standards with the existing criteria. As

for the second requirement, align the model's relevance to specific security domains with the current criteria.

- **Step 3: Modify each criterion:** Tailor each criterion to reflect the unique requirements of the chosen domain. For example, based on the previous steps, criterion 1.4 can be customized to "Does the security maturity model conform to established security standards and guidelines such as ISO/IEC 27001?". Similarly, criterion 1.5 can be adapted to "Does the model clearly specify its relevance to particular security domains or areas, such as cybersecurity, information security, incident response, *etc.*,?"

- **Step 4: Validate the customization:** Ensure that each customized criterion accurately reflects the unique aspects of the chosen domain.

Following these steps, the framework can be customized for any specific software domain, ensuring the criteria are relevant, comprehensive, and effective in assessing maturity models within the chosen context.

## CASE STUDY

In order to demonstrate the practicality, usability, and effectiveness of our framework in assessing maturity models, we conducted four detailed case studies to ensure comprehensive framework validation. We chose models from various software engineering domains to highlight our framework's cross-domain applicability. We aimed to select models developed in the last 5 years, as these will better indicate current practices in the field. We also tried to choose models with the authors available and willing to participate in the assessment process. Accordingly, we invited academic experts who had developed relevant maturity models to contribute to our study. Four researchers agreed to participate, allowing us to evaluate maturity models across various domains, including software outsourcing, security, integration, and sustainable software development. The case studies were each undertaken by a single experienced researcher. The maturity models are called maturity models A, B, C, and D to maintain confidentiality. The structured assessment process, aligned with ISO/IEC 15504-3 protocols, revealed distinct patterns of strengths and areas for improvement across all models. All models scored highly overall, placing them in the 'Advanced' category. Appendix A summarizes the assessment results for each of the evaluated maturity models. Below, we will discuss their specific scores and percentages to provide deeper insights into their unique capabilities and limitations.

### Case study results

#### *Maturity model A*

Maturity model A provides a framework to assess, manage, and improve security practices throughout the software development lifecycle, ensuring that security considerations are integral to the development processes. It comprises seven security assurance levels: governance and security threat analysis, secure requirement analysis, secure design, secure coding, secure testing and review, secure deployment, and security improvement. The levels consist of 46 critical security risks with 388 corresponding practices. This maturity model was developed by examining previous well-known development models and

carrying out a systematic literature review. It was later validated through practical implementation in real-world settings.

Maturity model A offers an approach for improving software security practices with a final score of 66, or 86.8%. The model's high score of (17 out of 18) in the Basic Information category indicates that it performs generally well in terms of validation and compliance. This implies that the model complies with existing security standards and has been verified in various real-world settings. The model's robust structure, scoring 19 out of 20 in the Model Structure category, also suggests that security levels and expected outcomes are well-defined with clear criteria and logical progression between levels. Additionally, it reveals the model's flexibility, allowing organizations to tailor the model to their specific needs. However, the lower scores in the Model Support (15 out of 20) and Model Assessment categories (15 out of 18) pose some limitations on the maturity model's overall quality. This might indicate the need for more refined assessment methodologies and enhanced support mechanisms in order to enhance applicability and ease of implementation.

### Maturity model B

Maturity model B assists vendor organizations in managing and executing outsourcing contracts effectively. This is done at various stages of the outsourcing contract life cycle: precontract, during contract, and post-contract. In order to implement the model successfully, nine critical success factors (CSFs) and ten barriers (CBs) affecting contract management were identified. Additionally, practical guidelines were provided for successfully identifying and resolving barriers. A systematic review of the literature and empirical studies led to the development of this maturity model.

Maturity model B scored 55, equivalent to 83.3%, with some criteria marked as not applicable. This model's strengths lie in its well-defined structure, scoring 15 out of 16 in the Model Structure category, making it practical for its intended use. However, its assessment methods, which scored (14 out of 18), are inadequate for assessing maturity effectively. Looking closely at this category, the assessment instruments were not validated, and there were no clear, precise criteria for assessing maturity at each level and dimension. In addition, the model does not support different types of assessments, nor does it outline clear procedures for the assessors. Addressing these areas would enhance the model's overall quality. On the other hand, the Model's Support category received (13 out of 14), demonstrating the model's strength in this area. It provides comprehensive support tools, including documentation and training materials for the intended users. Similarly, the basic information category scored (13 out of 18), suggesting that the model could have been enhanced with further validation and a clear specification of its relevance to particular domains or areas, ultimately increasing its credibility and applicability.

### Maturity model C

Maturity model C was created to help vendors measure their agile maturity for green and sustainable software development. This maturity model intends to address an emerging need in the software industry to develop environmentally sustainable software using

limited energy and resources. The model consists of several modules that measure different dimensions of agile maturity, including critical success factors (CSFs) and critical risk factors (CRFs) that may help or hinder the implementation of green software engineering. Maturity model C was built using existing models such as the Capability Maturity Model Integration (CMMI), Integrated Maturity Model (IMM), and the Software Operational Viability Risk Model (SOVRM). This maturity model has levels and criteria to assess organizations' practices, processes, and capabilities to achieve sustainability goals using agile methodologies.

Maturity model C achieved an overall score of 69, translating to 93.2%. This demonstrates that the model performs well in assessing sustainability integration into agile software development practices. Looking at the model's scores across the four categories, we can observe that the model's strengths lie in its well-organized structure and assessment capabilities. The model scored (18 out of 20) and (17 out of 18) in the Model Structure and Assessment categories, respectively. This indicates that the model has a well-defined structure with clear maturity levels and evaluation criteria to enhance green software development. However, the model scored lower in the Support category (16 out of 20), showing that there is a need for tailored support mechanisms, more documentation, and to enhance the model's accessibility and usability in different organizational settings.

### Maturity model D

Maturity model D was developed to overcome many of the issues associated with the integration of software components developed by distributed teams. This maturity model is intended to provide vendors with a systematic way to address integration issues in many different types of projects and products. The maturity model was developed from an extensive literature review, which identified critical elements and issues associated with software integration; this was then validated through an empirical study involving industry practitioners.

Maturity model D achieved a total score of 69, resulting in a percentage of 91%. This maturity model was particularly strong in its structural clarity, scoring 20 out of 20, the highest among all evaluated maturity models. This reflects its well-organized structure, including the clarity of the maturity model's levels and attributes. The maturity model scored (17 out of 18) in the model assessment category, indicating that the maturity model was effective in evaluating the integration practices with its current assessment methods. Maturity model D performs well in general but demonstrates weakness in the model support category, scoring 15 out of 20. This could indicate there are areas for improvement regarding documentation, customization, and plans for evolution and maintenance.

## DISCUSSION

The case studies revealed detailed insights about each model's quality and implementation readiness. The evaluation results showed that models A and C scored highly overall, demonstrating strong theoretical foundations and clear domain specifications. These

models particularly excelled in criteria related to established industry standards and research design methodology. Similarly, model D's results were high (17 out of 18), showing slight limitations in industry standards conformance. Model B's lowest score (13 out of 18) highlighted gaps primarily in validation evidence and domain relevance. Nevertheless, all models successfully followed predefined research steps and maintained clear ideological foundations.

The Model Structure category emerged as the strongest category with relatively high scores across all models. Model D scored (20 out of 20), demonstrating clarity in maturity levels and progression paths. Model A followed closely (19 out of 20) with only minor limitations regarding the framework's ease of use, without prior training being needed. Models C (18 out of 20) and B (15 out of 16) demonstrated well-defined structures but showed some constraints in adaptability and ease of use. The high scores in this category indicate mature approaches to model architecture and design.

The Model Assessment category's criteria revealed important insights. Models C and D (17 out of 18) demonstrated robust assessment procedures and clear evaluation criteria. On the other hand, Model A scored lower (15 out of 18) due to limitations in assessment method variety, lack of support, and misalignment between the model's design and the chosen assessment methods. Additionally, model B's score (14 out of 18) indicated a need to validate the assessment instrument and to enhance the assessment criteria's clarity.

Finally, all models achieved lower scores in the Support category. Model C scored the highest (16 out of 20), followed by models A and D, both scoring 15 out of 20, and model B (13 out of 14, with several inapplicable criteria). Common limitations among all the assessed models included insufficient documentation, limited benchmarking capabilities, the lack of support during the assessment process, and incomplete evolution planning. However, all models demonstrated strength in results communication and improvement recommendations.

These findings emphasize that, despite strong structural foundations and theoretical clarity, practical limitations such as missing support mechanisms, undefined long-term planning, and partial inapplicability hinder the quality of maturity models. The comparative analysis confirms that our measurement framework successfully identifies strengths and weaknesses across various software maturity models. Structural design and theoretical foundations were identified as key strengths, but significant deficiencies are evident in areas such as implementation support and assessment rigor. The framework's ability to generate actionable insights for model improvement across all case studies provides pragmatic evidence for construct validity as it identifies improvement opportunities that align with practitioner needs. The consistent evaluation across diverse domains demonstrates the framework's utility as a standardized assessment method. Although validation efforts concentrated on academically developed models, the framework's systematic methodology and comprehensive criteria indicate its promising applicability within industrial contexts.

## Threats to validity

Although our assessment framework was developed using a rigorous methodology, it is important to acknowledge several threats to its validity:

- Construct validity: The criteria used in the framework are derived mainly from the literature. However, there may be discrepancies between these criteria and other factors influencing the quality of software maturity models. Although we tried to ensure that our criteria comprehensively covered all relevant aspects of software maturity models, we may have overlooked research articles containing additional important criteria. Another construct validity concern is whether the framework truly measures maturity model quality. Even though we did not conduct statistical or qualitative testing for construct validity, the design of our approach supports the framework's validity. The criteria were derived from a literature review, ensuring alignment with established quality indicators for maturity models. They were further validated by domain experts to confirm their relevance and coverage of key quality dimensions. Additionally, case studies were utilized to measure the framework's practical utility in assessing different models and identifying meaningful areas for improvement, while aligning with ISO/IEC 15504-3 as a theoretical framework.

- Internal validity: The validation of our framework involved structured expert evaluations from industry practitioners and academics. Although their insights are valuable, potential bias may have influenced their responses due to personal experiences and perspectives. Moreover, in the case studies, assessors were required to evaluate their own developed maturity models. This might introduce the risk of bias, as they may unintentionally evaluate their maturity models positively. To mitigate this threat, we provided assessment guidelines from ISO/IEC 15504-3 standards to ensure reliable and comparable assessment results.

- Reliability: The consistency of the assessment using our framework could be affected by the subjective interpretation of the criteria by different assessors. Although some subjectivity is inherent in qualitative assessments, training and clear guidelines will be provided to mitigate this threat.

## CONCLUSION AND FUTURE WORK

This article proposes an assessment framework that assesses the quality of software maturity models. We started with a literature review and used our experience with software maturity models to develop the framework with criteria that measure software maturity models's quality. We integrated ISO/IEC 15504-3 assessment guidelines to enhance our framework further, and we validated the framework through expert reviews and case studies to ensure it is practical and works well in real-world applications. The framework covers all the aspects to be considered when developing software maturity models. It is organized into four main categories: basic information of the model, model structure, model assessment, and model support mechanisms. We aligned our work with ISO/IEC TR 15504-3 standards to ensure the assessment is consistent and reliable. Our framework provides valuable insights for the software industry and can be used by

professionals and academics to assess and improve maturity model development practices. Our developed framework demonstrates the need for a systematic approach to assess and evaluate software maturity models. Furthermore, feedback from experts indicated that the framework can drive improvements in the development of software maturity models.

This framework represents a structured approach to maturity model assessment and acknowledges several limitations. The purposive sampling for case studies and the inherent subjectivity in qualitative assessments represent areas for future improvement. In addition, our case studies focus primarily on academically developed models, which provide comprehensive documentation and developer accessibility necessary for thorough evaluation but may not fully represent the diversity of industry-developed proprietary models. Future studies should explore partnerships with industry organizations to validate framework applicability across proprietary models with appropriate measures in place to maintain confidentiality.

**Appendix A  Assessment results summary for the evaluated maturity models.**

| | | Score MM (A) | MM (B) | MM (C) | MM (D) |
|---|---|---|---|---|---|
| **1 Model basic information** | | | | | |
| 1.1 | Are the costs of implementing this maturity model (e.g., initial, implementation, recurring fees) reasonable relative to the value of applying it? | 2 | 1 | 2 | 2 |
| 1.2 | Were predefined steps followed in the research design of the maturity model? | 2 | 2 | 2 | 2 |
| 1.3 | Is the maturity model distinct and unique compared to existing models in the same domain? | 2 | 2 | 2 | 2 |
| 1.4 | Does the maturity model conform to established industry standards or guidelines? | 2 | 2 | 2 | 1 |
| 1.5 | Does the model clearly specify its relevance to particular domains or areas? | 2 | 1 | 2 | 2 |
| 1.6 | Has the model been validated in real-world settings (e.g., peer-reviewed literature, surveys, industry groups), demonstrating its applicability and effectiveness? | 2 | 1 | 2 | 2 |
| 1.7 | Does the model have a clear ideological foundation supported by established theories or models? | 2 | 2 | 2 | 2 |
| 1.8 | Is the model evidence-based (e.g., grounded in the peer-reviewed literature, industry-recognized best practice)? | 1 | 1 | 2 | 2 |
| 1.9 | Are the model's practices applicable across different scenarios, cases, and projects? | 2 | 1 | 2 | 2 |
| **SECTION SUBTOTAL** | | 17 | 13 | 18 | 17 |
| **2 Model structure criteria** | | | | | |
| 2.1 | Is the process of applying the model clear? | 2 | 2 | 2 | 2 |
| 2.2 | Does the model provide clear definitions of maturity and dimensions of maturity? | 2 | 2 | 2 | 2 |
| 2.3 | Are maturity levels within the model clearly defined, with each level described by clear criteria and expected outcomes? | 2 | 2 | 2 | 2 |
| 2.4 | Does the maturity model outline specific levels and the logical progression between these levels? | 2 | NA | 2 | 2 |
| 2.5 | Is the maturity model's structure, including the number of levels and attributes, clear and coherent? | 2 | 2 | 2 | 2 |
| 2.6 | Does the model propose specific improvement measures or practices for advancing from one maturity level to the next? | 2 | 2 | 2 | 2 |
| 2.7 | Is there an ability to adjust or alter the model's structure, components, or processes (e.g., the model can evolve and remain relevant.)? | 2 | NA | 1 | 2 |

(Continued)

**Appendix A (continued)**

| | | Score MM (A) | MM (B) | MM (C) | MM (D) |
|---|---|---|---|---|---|
| 2.8 | Is there a balance in the model between reflecting the complexities of the domain and maintaining simplicity for understandability? | 2 | 2 | 2 | 2 |
| 2.9 | Is the maturity model's constructs and definitions accurate and precise? | 2 | 2 | 2 | 2 |
| 2.10 | Is the maturity model easily accessible and usable by practitioners without extensive training? | 1 | 1 | 1 | 2 |
| **SECTION SUBTOTAL** | | **19** | **15** | **18** | **20** |
| **3 Model assessment criteria** | | | | | |
| 3.1 | Were the model's assessment instruments validated to ensure accuracy and reliability? | 2 | 1 | 2 | 2 |
| 3.2 | Are there clear, precise criteria for assessing maturity at each level and dimension, allowing for consistent and objective evaluations? | 2 | 1 | 2 | 2 |
| 3.3 | Does the model include a detailed methodology for conducting assessments, providing guidance on evaluating criteria, and interpreting results? | 2 | 2 | 2 | 2 |
| 3.4 | Does the assessment methodology outline clear procedures for assessors? | 2 | 1 | 2 | 2 |
| 3.5 | Is there a logical connection between the model's design and the chosen assessment methods? | 1 | 2 | 2 | 2 |
| 3.6 | Does the model support different types of assessments (e.g., self-assessment, third-party assessment)? | 1 | 1 | 2 | 2 |
| 3.7 | Can support be provided during the assessment using the model? | 1 | 2 | 2 | 1 |
| 3.8 | Does the maturity model promote transparency and openness in identifying and addressing areas for improvement, including the possibility of suggesting enhancements? | 2 | 2 | 2 | 2 |
| 3.9 | Does the maturity model leverage technology and tools for more efficient and accurate assessments? | 2 | 2 | 1 | 2 |
| **SECTION SUBTOTAL** | | 15 | 14 | 17 | 17 |
| **4 Model support criteria** | | | | | |
| 4.1 | Does the maturity report communicate results clearly? | 2 | 2 | 2 | 2 |
| 4.2 | Is there adequate documentation supporting the application of the assessment, such as a handbook, textual descriptions, or software assessment tools? | 1 | NA | 1 | 1 |
| 4.3 | Is the model designed with enough flexibility to be adapted to different organizational settings? | 2 | 2 | 1 | 2 |
| 4.4 | Does the model provide actionable insights and guidance for both practitioners and researchers? | 2 | 2 | 2 | 2 |
| 4.5 | Does the maturity report provide practical, useful recommendations to drive improvements? | 2 | 2 | 2 | 2 |
| 4.6 | Does the model facilitate benchmarking against industry standards or comparisons with similar organizations? | 1 | 1 | 2 | 1 |
| 4.7 | Can the maturity report be customized? | 1 | NA | 2 | 1 |
| 4.8 | Is training available for effectively implementing and utilizing the maturity model? | 1 | NA | 2 | 2 |
| 4.9 | Is there a continuity and evolution plan between different versions of the model with accessible documentation? | 1 | 2 | U | 1 |
| 4.10 | Is there a maintenance plan in place to ensure the model remains relevant and up-to-date? | 2 | 2 | 2 | 1 |
| **SECTION SUBTOTAL** | | 15 | 13 | 16 | 15 |
| **Overall total** | | 66 | 55 | 69 | 69 |
| **Percentage** | | 86.8% | 83.3% | 93.2% | 91% |

### Funding

The authors received support provided by the Deanship of Research at King Fahd University of Petroleum and Minerals, Saudi Arabia. This work was funded by the Interdisciplinary Research Center for Intelligent Secure Systems under Grant #: INSS2404.

The funders had no role in study design, data collection and analysis, decision to publish, or preparation of the manuscript.

## Grant Disclosures

The following grant information was disclosed by the authors:
Deanship of Research at King Fahd University of Petroleum and Minerals, Saudi Arabia.
Interdisciplinary Research Center for Intelligent Secure Systems: INSS2404.

## Competing Interests

The authors declare that they have no competing interests.

## Author Contributions

- Reem Alshareef performed the experiments, analyzed the data, performed the computation work, prepared figures and/or tables, authored or reviewed drafts of the article, and approved the final draft.
- Mohammad Alshayeb conceived and designed the experiments, analyzed the data, performed the computation work, authored or reviewed drafts of the article, and approved the final draft.
- Mahmood Niazi conceived and designed the experiments, analyzed the data, authored or reviewed drafts of the article, and approved the final draft.
- Sajjad Mahmood conceived and designed the experiments, analyzed the data, authored or reviewed drafts of the article, and approved the final draft.

## Data Availability

The process automation and supplementary material is available at Zenodo: Alshareef, R. (2025). Measurement framework for software maturity models survey. Zenodo. https://doi.org/10.5281/zenodo.15834005.

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
