# Peer review of "A measurement framework to assess software maturity models"

_PeerJ Computer Science, doi:10.7717/peerj-cs.3183_

## Round 0.1 · original submission · Major Revisions

Both reviewers welcome this work, and complement your efforts in aproaching this complex problem. However, they also highlight specific issues with the manuscript's structure, grammar and typography that should be addressed. They also note weakness in study design and lack of detail in reporting, such as explaining how the case study participants were selected, which participants assessed which model (or whether all participants assessed all models and results were averaged in some way).

Please address their points as fully as possible, and revise the manuscript accordingly.

Where reviewers suggest more case study work to be undertaken it is acceptable to justify whether this could not be done, but it is important to provide additional detail where possible - particulary if more clarity regarding participant's biases due to their involvement in the development of particular software maturity models.

The provision of the Excel spreadsheet version of the tool at http://softwareengineeringresearch.net/MaturityModelAssessment.xlsx is welcome, but this URL should be 'https' - and ideally the questionnaire should be deposited at Zenodo or figshare in accordance with PeerJ CS's supplementary materials and open data policies.


**Language Note:** The review process has identified that the English language must be improved. PeerJ can provide language editing services - please contact us at [email protected] for pricing (be sure to provide your manuscript number and title). Alternatively, you should make your own arrangements to improve the language quality and provide details in your response letter. – PeerJ Staff

·

Basic reporting

1. The article lacks consistency in formatting, for example, the font used in Section 2 (lines 128–132) differs from the rest of the manuscript, and the first paragraph in the Introduction is in justified style, which affects readability.
2. There are typographical errors, such as the word "Appenidx" on line 529, which should be corrected to "Appendix." Please review the entire content of the article thoroughly.
3. The structure of the sections is disorganized, as Section 7 is listed twice (lines 122–123), which may cause confusion for the reader. There should be nine sections, with Section 8 listing the threats to validity.
4. Figure 2 lacks sufficient explanation. The metrics presented are not described, and various syntax elements, such as color codes and borderlines, are not explained. A legend should be added to help readers understand the meaning behind these visual cues.
5. The references need improvement in both quality and clarity. Many cited papers are more than 10 years old, and some references lack complete publication information. For instance, references [15] and [12] do not include the journal or proceedings where the papers were published.

Experimental design

1. The rest of this paper is not clearly reflected in the structure of the article, making it difficult to follow the sequence of the research steps.
2. The sources of the literature are not clearly identified in terms of whether they are from existing academic literature, research papers, or industry reports, nor is the composition of these sources explained. Additionally, there is no supporting data indicating the quality of the cited works.
3. Please explain the reason for selecting ISO/IEC TR 15504-3 as the standard guideline. Please also provide a clear explanation of its structure!
4. On what basis are the groupings into four categories (lines 316–318) made, and how is the score interpretation in lines 359–365 determined?
5. Please specify which version of CMMI is used as the reference in this study.

Validity of the findings

1. Lines 458–459 mention the involvement of five domain experts; however, Table 8 shows only five individuals with three different roles. Please clarify this inconsistency.
2. The case study mentions the involvement of four researchers, but the data in Appendix A appear to be rounded. Was rounding applied? If so, what rounding method was used? It is recommended to include the raw responses from all researchers for transparency.

Additional comments

I highly appreciate the authors' research efforts; the topic presented is very interesting. However, several improvements are still needed to enhance the overall quality of the paper. Please revise based on the comments.

Cite this review as

Reviewer 2 ·

Basic reporting

The manuscript is generally well-written, logically structured, and appropriately formal for an academic audience. The progression from problem definition through literature review, framework development, validation, and conclusion is coherent and easy to follow. Key terms are used effectively, and the structure supports comprehension.

However, there are areas where the writing could be improved for clarity and precision. Some sentences are overly long or awkwardly constructed.
For example, "To maintain the model’s consistency, reliability, and scalability over time, the criteria included design clarity, standardization, accuracy, and documentation. Additionally, they encompass aspects of the maturity model’s effectiveness, practicality, validation, support, and application."

There are occasional inconsistencies in verb tense usage and minor grammatical issues that could distract the reader.
For example, "We wanted to capture as many criteria as possible across scalability, empirical evidence, and usability in practice. These themes kept appearing in the literature as key to a high-quality maturity model." The first sentence uses past tense ("wanted to capture"), while the second sentence continues with past progressive ("kept appearing"), creating a slightly disjointed timeline. Since this reflects general findings from the literature, the present tense may be more appropriate for consistency and academic tone.
"Each model scored highly overall, placing them in the 'Advanced' category." The subject “each model” is singular, but the pronoun “them” is plural, creating a pronoun agreement error.

Experimental design

The methodology presented in this paper is well-structured, logically phased, and demonstrates a commendable balance between theoretical grounding and empirical validation. The five-phase approach, including literature review, framework development, integration of ISO/IEC 15504-3, expert evaluation, case studies, and documentation, reflects a thorough and pragmatic research design. The integration of international standards and the use of expert feedback and case studies from multiple software domains enhances the framework’s relevance and practical applicability.

However, several areas could be strengthened to improve the methodological rigor.
Firstly, while the paper reports quality scores from case studies, the scoring mechanism lacks sufficient detail. It is unclear how individual criteria were weighted, how scores were derived, and whether inter-rater reliability was assessed. Including such quantitative validation metrics would significantly enhance the framework’s reliability and replicability.

Secondly, the process for selecting maturity models used in the case studies should be clarified. As it stands, there is a potential risk of selection bias that could affect generalizability. Additionally, the framework evaluation process is a commendable attempt to validate usability, structure, and applicability through expert feedback. However, the limited sample size (five experts), the short evaluation duration, and the lack of detailed methodological description weaken the rigor and generalizability of the findings. The paper would benefit from a more transparent account of the evaluation process, including item-level metrics and qualitative analysis of expert comments. Additionally, expanding the expert pool to include more industry practitioners would enhance the framework’s practical credibility.

The inclusion of four detailed case studies from diverse software domains (outsourcing, security, integration, and sustainable development) is a clear strength of this study. It demonstrates the framework’s practical utility and adaptability across varied maturity model contexts. Collaborating directly with the model developers adds further authenticity to the evaluation process, while the use of structured assessment protocols aligned with ISO/IEC 15504-3 reflects a disciplined and ethically sound methodology.

However, the case study selection process warrants further clarification and improvement. The reliance on invitation-based participation constitutes a form of convenience sampling, which introduces potential selection bias. There is also no information on the criteria used to identify or invite participants, making it difficult to assess the representativeness or diversity of the sample beyond domain type. Additionally, the sample size, while sufficient for preliminary insights, is too limited to support broader generalizations about the framework’s robustness.
To enhance the study's credibility, I recommend that the authors:
• Clearly articulate the selection criteria for both participants and maturity models.
• Justify the sample size and its adequacy relative to the framework’s validation goals.
• Consider incorporating a structured cross-case comparative analysis to identify consistent strengths or shortcomings across models.
• Explore opportunities to expand the sample or include models from industry, where possible.

Validity of the findings

The framework development process is well-grounded in literature and established models such as ISO/IEC (SPICE) and CMMI. The clear categorization into four key dimensions, Model Basic Information, Model Structure, Model Assessment, and Model Support, demonstrates logical structure and comprehensive coverage of both technical and practical aspects of maturity models. The inclusion of scalability, empirical validation, and usability enhances the framework's practical relevance.

However, several areas could be improved:
1. Insufficient Methodological Transparency
While the framework is said to be derived from literature and the authors’ prior work, the process lacks systematic criteria elicitation methods. It is unclear how many studies were reviewed, how sources were selected, or whether a formalized synthesis method (e.g., thematic analysis or systematic review protocol) was used to extract and categorize criteria. This weakens the traceability and reproducibility of the framework.

2. Lack of Weighting or Prioritization of Criteria
The framework presents numerous criteria across four categories, but does not discuss how these are weighted or prioritized. Without a scoring rubric, decision rules, or differentiation of criteria importance, the assessment risks becoming overly subjective and inconsistent across use cases. This is particularly critical for organizations seeking to make informed comparisons between models.

4. Limited Empirical Grounding During Development
Although empirical insights were later incorporated through expert evaluation and case studies during validation, the initial framework development phase appears largely conceptual. Including proper empirical feedback earlier in the criteria formulation process, such as through pilot testing with practitioners or feedback on draft criteria, could have ensured that the framework better reflects real-world challenges and use contexts from the outset.

5. No Explicit Validation of Construct Validity
There is no mention of testing the construct validity of the framework (e.g., does it truly measure maturity model quality?), either statistically or qualitatively. Without this, it's unclear whether the chosen dimensions adequately capture all relevant aspects of maturity model performance or success.

The conclusion of the paper effectively highlights the key contributions of the paper, including the development and validation of the software maturity model assessment framework. The incorporation of ISO/IEC 15504-3 standards and the expert feedback validation are well summarized, showcasing the framework's practical relevance.

However, the conclusion would benefit from a more critical reflection on the study's limitations, such as the small expert sample size, potential biases in the case study selection, and the inherent subjectivity of the qualitative assessment methods used. A more balanced assessment of these limitations would strengthen the paper's credibility.

Cite this review as

---

## Round 0.2 · accepted · Accept

Both reviewers welcomed your revised version of the manuscript, and whilst one notes an inconsistency in typographic style in one reference, I recommend this can be addressed at the proof stage.

It would of course also be ideal if a more recent and diverse set of literature were considered (as noted by the same reviewer), but I recognise that this would entail significant additional work. Thanks once again for your submission, and I look forward to seeing the larger community of software maturity researchers build upon the proposed model.

·

Basic reporting

The authors have addressed 4 out of the five revision points I previously mentioned:
1. The article format is now consistent.
2. There are no longer any typographical errors.
3. The article structure is well-organized and easy to read.
4. An explanation for Figure 2 has been added.
However, for the final point regarding references, the author has not fully addressed the revision. Although some improvements have been made to complete the citation details, it is worth noting that 29 out of 39 cited papers are more than 10 years old. Furthermore, the reference formatting for item [34] still does not follow the standard style.

Experimental design

The authors have revised based on the comments I provided:
1. The article is written in accordance with the sequence of the research steps.
2. The literature data is clearly presented.
3. The rationale for selecting ISO/IEC TR 15504-3 has been adequately explained.
4. The basis for the grouping of categories has been clarified.
5. The author employs CMMI 2.0 in the study.

Validity of the findings

The author has addressed both revision points:
1. The number of experts has been increased from 5 to 11, along with a clear explanation.
2. Appendix A has been properly clarified.

Additional comments

I sincerely appreciate the authors' efforts in revising the manuscript based on my comments.

Cite this review as

Reviewer 2 ·

Basic reporting

The manuscript is well-written, logically organized, and suitably formal for an academic audience. It presents a clear and coherent progression from problem definition through literature review, framework development, validation, and conclusion. Key terms are applied consistently, and the overall structure enhances clarity and comprehension.

Experimental design

The methodology outlined in this paper is well-structured, logically sequenced, and achieves an effective balance between theoretical foundation and empirical validation. The five-phase approach, comprising literature review, framework development, integration of ISO/IEC 15504-3, expert evaluation, case studies, and documentation demonstrates a comprehensive and pragmatic research design. Incorporating international standards, along with expert feedback and case studies from diverse software domains, further strengthens the framework’s relevance and practical applicability.

Validity of the findings

The framework development process is firmly grounded in literature and established models such as ISO/IEC (SPICE) and CMMI. Its clear categorization into four key dimensions, Model Basic Information, Model Structure, Model Assessment, and Model Support reflects a logical structure and comprehensive coverage of both the technical and practical aspects of maturity models. The incorporation of scalability, empirical validation, and usability further enhances its practical relevance.

Cite this review as